# Development and Validation of Artificial-Intelligence-Based Radiomics Model Using Computed Tomography Features for Preoperative Risk Stratification of Gastrointestinal Stromal Tumors

**DOI:** 10.3390/jpm13050717

**Published:** 2023-04-24

**Authors:** Marco Rengo, Alessandro Onori, Damiano Caruso, Davide Bellini, Francesco Carbonetti, Domenico De Santis, Simone Vicini, Marta Zerunian, Elsa Iannicelli, Iacopo Carbone, Andrea Laghi

**Affiliations:** 1Department of Medical-Surgical Sciences and Biotechnologies, Academic Diagnostic Imaging Division, I.C.O.T. Hospital, University of Rome Sapienza, Via F. Faggiana 1668, 04100 Latina, Italy; 2Department of Radiological, Oncological and Pathological Sciences, Academic Diagnostic Imaging Division, I.C.O.T. Hospital, University of Rome Sapienza, Via F. Faggiana 1668, 04100 Latina, Italy; 3Department of Surgical and Medical Sciences and Translational Medicine, Radiology Unit, Sant’Andrea University Hospital, University of Rome Sapienza, Via di Grottarossa 1035, 00189 Rome, Italy; 4Radiology Unit, Sant’Eugenio Hospital, Piazzale dell’Umanesimo 10, 00144 Rome, Italy

**Keywords:** gastrointestinal stromal tumor, radiomics, machine learning, risk assessment, prognostic, artificial intelligence, computed tomography

## Abstract

Background: preoperative risk assessment of gastrointestinal stromal tumors (GISTS) is required for optimal and personalized treatment planning. Radiomics features are promising tools to predict risk assessment. The purpose of this study is to develop and validate an artificial intelligence classification algorithm, based on CT features, to define GIST’s prognosis as determined by the Miettinen classification. Methods: patients with histological diagnosis of GIST and CT studies were retrospectively enrolled. Eight morphologic and 30 texture CT features were extracted from each tumor and combined to obtain three models (morphologic, texture and combined). Data were analyzed using a machine learning classification (WEKA). For each classification process, sensitivity, specificity, accuracy and area under the curve were evaluated. Inter- and intra-reader agreement were also calculated. Results: 52 patients were evaluated. In the validation population, highest performances were obtained by the combined model (SE 85.7%, SP 90.9%, ACC 88.8%, and AUC 0.954) followed by the morphologic (SE 66.6%, SP 81.8%, ACC 76.4%, and AUC 0.742) and texture (SE 50%, SP 72.7%, ACC 64.7%, and AUC 0.613) models. Reproducibility was high of all manual evaluations. Conclusions: the AI-based radiomics model using a CT feature demonstrates good predictive performance for preoperative risk stratification of GISTs.

## 1. Introduction

Gastrointestinal stromal tumors (GISTs) are the most common mesenchymal tumors found in the gastrointestinal tract, accounting for about 2% of gastrointestinal tumors, with an incidence that has been progressively increasing over the past year. [1,2].

These tumors are derived from precursors of interstitial Cajal cells, pacemaker cells responsible for (GI) peristalsis activity. Currently, no environmental risk factor for GIST is known, but there is evidence of familial predisposition to germline oncogene mutations: KIT or PDFRA oncogene mutations are the most frequent [3].

Unlike other tumors, for which the TNM system represents the most commonly adopted staging tool, the risk stratification of GISTs is based on the Miettinen’s classification, which has been recently reviewed [4]. By integrating tumor size (2 cm; >2–5 cm; >5–10 cm; >10 cm), mitotic index (5/50 HPFs or >5/50 HPFs), and tumor site (stomach; duodenum; small bowel; rectum), this classification identifies five risk grades: none, very low, low, moderate, and high. The prognosis of GISTs is closely related to their risk grade. Different risk grades lead to different therapeutical options. Therefore, an adequate preoperative tumor assessment, including specimen collection and pathological examination based on microscopic morphology and immune phenotype, is mandatory to select an optimal therapeutic strategy for each patient [5].

Multidetector computed tomography (MDCT) plays a key role in GIST management, including detection, evaluation of tumor extent, and evaluation of treatment response. However, less is known about its role in risk assessment and prognostication of GISTs [6,7].

Multiple MDCT findings are helpful to establish the preoperative risk stratification of GISTs; given that preoperative biopsy for histopathological assessment is not routinely performed due to the risk of bleeding and/or seeding of the tumor, MDCT imaging findings are helpful in the preoperative risk stratification of GISTs [8,9].

In this setting, the emerging roles of artificial intelligence (AI) and radiomics offer new opportunities to forecast the tumor risk and aid in clinical decision making [10,11].

In particular, texture analysis (TA) has been increasingly applied to radiological imaging for diagnosing, characterizing, and monitoring treatment response by quantifying tumor heterogeneity and irregularity of tissue components [12,13]. Tumors with high heterogeneity have been shown to have worse prognosis, potentially reflecting intrinsic biological aggressiveness or treatment resistance [14,15,16,17,18,19]. 

Recently, some studies investigated whether MDCT TA features of GISTs could be used as imaging biomarkers, demonstrating its potential role in the characterization of tumor subtypes [20,21,22,23]. In these studies investigators developed different methods to extract CT features using customized software or complex AI algorithms. This complexity may limit the clinical application of such promising algorithms.

Thus, the aim of our study was to develop and validate classification models based on morphologic and texture features extracted from CT images, to predict a tumor’s biology using the Miettinen’s classification as a reference standard.

## 2. Materials and Methods

### 2.1. Study Design and Population

This retrospective, non-randomized, single center study was conducted according to the Good Clinical Practice (GCP) International Conference on Harmonization (ICH). We retrospectively selected patients with a pathological diagnosis of GIST who had undergone a multiphasic CT scan of the abdomen from May 2017 to September 2019. Inclusion criteria were (1) histopathological diagnosis of GIST, and (2) surgical excision of the tumor. Exclusion criteria included (1) a poor image quality of the CT images, (2) an incomplete histopathological report, and (3) neoadjuvant chemotherapy before CT.

The study was approved by the local ethical committee. Informed consent was waived because of the retrospective nature of the study and the anonymization of clinical data.

### 2.2. Pathological Examinations

The histopathological diagnosis of GIST was performed by an expert pathologist with more than 20 years of experience, based on microscopic morphology and immune-phenotype. Immunohistochemistry (IHC) was performed on freshly cut, 3-micron-thick, paraffin-embedded tissue sections using antibodies against C-KIT/CD117 (Dako A 4502, polyclonal rabbit anti-human), according to manufacturer instructions. All the included cases demonstrated cytoplasmatic/membranous positivity. Mitotic count was performed on 50 HPF (high power field) and expressed as the number of mitoses/50 high power fields (HPF). Tumor size was measured on formalin-fixed samples, and expressed in cm. As per Miettinen’s classification, tumors were stratified in five risk classes (no risk, very low risk, low risk, moderate risk, high risk) based on mitotic count, tumor size and location (Table 1) [4]. The five risk classes were dichotomized in two groups: a higher risk group (including moderate and high risk classes) and a lower risk group (including no risk, very low risk and low risk classes).

### 2.3. MDCT Acquisition Protocol

All MDCT scans were acquired with a 16 raw scanner (LighSpeed 16 slice, GE Medical Systems, Waukesha, WI, USA). All acquisitions were performed in the cranio-caudal direction form the diaphragmatic dome to the end of the ischiatic branches. Scanning parameters were as follows: kV 120; mAs 120–180; gantry rotation0.5 s; pitch 1:1; detector configuration 16 × 1.5 mm; reconstructed section thickness 2.5 mm; standard reconstruction algorithm. A portal venous phase, following unenhanced scan, was acquired after 75 s from the injection of 0.625 mL of iodine per Kg of total body weight injected at 1.6 gI/s.

### 2.4. Morphologic Features

For each tumor, the following features were evaluated by two independent radiologists (with more than 10 years of experience in abdominal radiology): primary tumor location, lesion margins, angiogenesis, intralesional necrosis, peritoneal effusion, peritoneal implants, degree and pattern of contrast enhancement, and invasion of adjacent organs. Radiologists were blinded to the histopathological outcome of the tumors.

The primary tumor location was classified according to the gastrointestinal tract of origin: esophagus, stomach, duodenum, jejunum, ileum, and colon. The margins of the lesions were classified as regular when the edge of the lesion appeared smooth, or irregular when they appeared jagged. The presence of angiogenesis was assessed when enlarged and engorged blood vessels, close to the lesion, were identified. The presence of internal necrosis was assessed when intratumoral low-attenuation unenhanced areas were identified. Both peritoneal effusion and implants were scored as present or absent. The density of the primary tumor was measured applying a circular ROI on unenhanced and portal venous phase images. The degree of contrast enhancement was scored as mild, in cases where the difference between enhanced density and unenhanced density was lower than 55.33 HU, and as high if it was greater or equal than the same value [24]. The enhancement pattern was classified as homogeneous or heterogeneous based on the presence of different attenuation areas within the tumor. Finally, the invasion of adjacent organs was defined as an absence of clear margins between the tumor and the adjacent structures.

### 2.5. Texture Features

Texture features were extracted using TexRAD, a proprietary software algorithm (TexRAD Ltd., London, UK) commercially available and equipped with a simple user interface. The feature extraction process was performed by two independent radiologists (with 10 years of experience in abdominal imaging), blinded to the histopathological results.

A region of interest (ROI) was drawn around the tumor at the level of the largest tumor area as depicted on the axial MDCT portal venous phase images. The ROI was then used for texture analysis, which comprised an image histogram technique with an initial image filtration, followed by the quantification of texture within the filtered images. The in-plane filtration step was performed by means of a Laplacian of Gaussian spatial band-pass filter to produce a series of derived images highlighting features at different spatial scaling factors (SSF), ranging from fine to coarse texture within an ROI. The scale was selected by altering the filter standard deviation parameter, or σ, between 0.0 (not filtered) and 2 (coarse texture); SSFs performed by the software were: 1 mm, 1.5 mm, 1.8 mm, and 2 mm. A value of 1 mm represented fine texture scale, a value of 1.5 mm and 1.8 mm represented medium texture scale, and 2 mm represented a coarse texture scale. Heterogeneity within each ROI was quantified with and without image filtration using the following histogram parameters: kurtosis, entropy, skewness, mean value of positive pixels (MPP), standard deviation (SD), and mean. Kurtosis, which can be positive or negative, reflects the peakedness of the histogram. Entropy is linked with the irregularity of gray-level distribution. Skewness represents and measures the asymmetry of the histogram and could be positive or negative. The mean is the average value of the pixels within the analyzed ROI. SD describes the variation, low or high, from the average (mean value). MPP represents the average brightness of positive pixel values within the image [25,26].

### 2.6. Machine Learning Classification

Both the morphologic and the texture features extracted from CT images were combined and analyzed using the WEKA (Waikato Environment for Knowledge Analysis, Version 3.8.5, University of Waikato, Hamilton, New Zealand) machine learning (ML) suite for data mining classification. A total of thirty-eight features were extracted: eight morphologic features and thirty texture features. The aim of this process was to identify a ML classification algorithm able to identify higher- and lower-risk patients as determined by the Miettinen’s classification, which was considered the reference standard.

During the first step, patients were subdivided in two groups. Using the WEKA Explorer Filter tool, two thirds of the patients were placed in the training group, and one third in the validation group, after the population had initially been randomly reorganized. The training group was analyzed using Auto-WEKA, a dedicated package that allows the automatic identification of the best model with its best parameter settings (hyperparameter optimization) for a given classification or regression task, as well as a feature selection process. This analysis was performed separately for the eight morphologic features and the 30 texture features. Finally, the process was performed on all 38 features merged.

The optimized classification algorithms, identified by the Auto-WEKA analysis, were applied to the validation group, performing three separate analyses: morphologic features, texture features, and combined (morphologic and texture) features. For each classification model, sensitivity (SE), specificity (SP), accuracy (ACC) and area under the curve (AUC) were evaluated.

### 2.7. Statistical Analysis

All continuous variables were expressed as mean and standard deviation (SD).

Differences in patients’ sex distribution, tumor location, tumor size, mitotic rate, Miettinen’s risk score and morphologic features were calculated using a χ^2^ test with Yates’s correction. The Student *t* test was calculated to find significant differences in patients’ age.

The one-way ANOVA with Fisher’s LSD test was used to find significant differences in texture features.

Since the features implemented in the classification algorithms were derived from manual assessments, intra-reader and inter-reader agreement were calculated. The reproducibility of the morphologic feature evaluation was calculated with the weighted Cohen’s kappa (κ) analysis, while the reproducibility of the texture feature measurement was calculated using the intraclass correlation coefficient (ICC). One of the two radiologist performed all measurements twice for intra-reader agreement. Agreement was interpreted according to the following criteria: >0.81: excellent agreement; 0.61–0.80: good agreement; 0.41–0.60: moderate agreement; 0.21–0.40: fair agreement; <0.20: poor agreement.

All statistical analyses were carried out using SPSS (Version 25.0. IBM Corp.: Armonk, NY, USA), GraphPad Prism version 7.0 (GraphPad Software, La Jolla, CA, USA) and MedCalc (MedCalc Software^®^ version 12.5, Ostend, Belgium).

A two-tailed *p* < 0.05 was considered statistically significant.

## 3. Results

### 3.1. Study Population

Eighty-one patients were retrospectively selected from our database. Twenty-nine patients were excluded from the analysis because of a incomplete histology report (17), low-quality CT images (4), or neoadjuvant chemotherapy before CT (8). Thus, the final study population resulted in fifty-two patients (Figure 1).

At histology, 25% (13) of the included patients presented more than five mitoses (from 6 to 180), and 75% (39) five or less (from 0 to 5). As for tumor location, 71.2% (37) of the tumors were located in the stomach, 9.6% (5) in the duodenum, 7.7% (4) in the jejunum, 9.6% (5) in the ileum, and 1.9% (1) in the esophagus. Three lesions (5.8%) were smaller than 2 cm, 25 lesions (48.1%) ranged between 2 and 5 cm, 11 lesions (21.1%) ranged between 5 and 10 cm, while 13 lesions (25%) were larger than 10 cm.

As per the reviewed Miettinen’s classification, lesions were stratified as follows: 5.8% (3) no risk, 27% (14) very low risk, 25% (13) low risk, 21.1% (11) moderate risk, and 21.1% (11) high risk. Accordingly, 22 patients (42.3%) were included in the higher risk group, and 30 patients (57.7%) in the lower risk group. No statistically significant differences were observed between the higher and lower risk groups in terms of gender, age, tumor location and mitotic rate, while a significant difference was observed in tumor size. Results are summarized in Table 2. 

The WEKA Explorer Filter tool randomly subdivided the population in two groups: the first group, including 31 patients (59.6%), used for model training, and the second group, made up of 21 patients (40.4%), for model testing. No statistically significant differences were observed between the training and the validation groups for any of the characteristics evaluated. Results are summarized in Table 2. 

### 3.2. Morphologic and Texture Features

The higher and lower risk groups differed significantly for most of the morphologic features (margins, angiogenesis, necrosis, peritoneal effusion, peritoneal seeding, organ invasion, and enhancement pattern) and some of the texture features (SF0mean, SF0MPP, SF1.5SD, SF1.5MPP, SF1.8mean, SF1.8SD, SF1.8MPP, SF2mean, SF2SD, and SF2MPP). No differences were observed between the training and validation groups for both morphologic and texture features. Results are summarized in Table 3 and Table 4.

### 3.3. Machine Learning Models Training

Among the 38 features, only 16 were selected for the development of the classification model according to the Auto-WEKA analysis (10-fold cross validation attribute evaluator “CorrelationAttributeEval”). Among the eight morphologic features, four of these were selected, including angiogenesis, necrosis, delta density, and enhancement pattern. On the other hand, 12 of the 30 texture features were selected for the model development: SF0_sd, SF0_entropy, SF0_skewnwss, SF1_mean, SF1_sd, SF1_entropy, SF15_mean, SF15_sd, SF15_entropy, SF18_mean, SF2_mean and SF2_sd.

Three models were subsequently developed: the first one based on morphologic features only (morphologic model), the second one on texture features only (texture model), and the last one using both feature classes (combined model).

The Multilayer Perceptron (MLP) classifier was identified as the best for the morphologic model. The hyperparameters were optimized as follows: *-L, 0.8440810869039, -M, 0.9072704953488756, -H, t, -S, 1*. The model diagnostic performance estimates were SE 92.3%, SP 81.8%, ACC 85.7%, and AUC 0.848.

The Locally Weighted Learning (LWL) classifier was identified as the best for the texture model. The hyperparameters were optimized as follows: *-U 0 -K −1 -A „weka.core.neighboursearch.LinearNNSearch -A \“weka.core.EuclideanDistance -R first-last\”„ -W weka.classifiers.trees.DecisionStump*. The diagnostic performance estimates were SE 75%, SP 93.3%, ACC 82.8%, and AUC 0.837.

The Multilayer Perceptron (MLP) classifier was identified as the best for the combined model. The hyperparameters were optimized as follows: *-L 0.3 -M 0.2 -N 500 -V 0 -S 0 -E 20 -H a*. The model performances were SE 100%, SP 95%, ACC 97.1%, and AUC 0.968.

Results are summarized in Table 5.

### 3.4. Machine Learning Models Validation

The three models were subsequently applied to the validation population. Just like the training models, the highest performances were obtained by the combined model (SE 85.7%, SP 90.9%, ACC 88.8%, and AUC 0.954) followed by the morphologic (SE 66.6%, SP 81.8%, ACC 76.4%, and AUC 0.742) and texture (SE 50%, SP 72.7%, ACC 64.7%, and AUC 0.613) models. Results are summarized in Table 5.

### 3.5. Reproducibility

The eight morphologic features and thirty texture features showed good or excellent agreement for both inter- and intra-reader evaluation. As for the morphologic features, the highest inter-reader agreement was achieved for lesion margins (κ = 0.95; 95% CI: 0.89–0.99), the lowest for enhancement pattern (κ = 0.91; 95% CI: 0.81–0.98). The highest intra-reader agreement was achieved for lesion margins (κ = 0.96; 95% CI: 0.91–0.99), while the lowest was observed for enhancement degree (κ = 0.93; 95% CI: 0.59–0.94). The intraclass correlation coefficients for the texture features were 0.79 (95% CI: 0.53–0.93) and 0.82 (95% CI: 0.59–0.94) for the inter- and intra-reader agreement, respectively.

## 4. Discussion

The aim of this study was to develop and validate a decision support model, based on the combination of CT morphologic and texture features, to classify patients affected by GIST as higher or lower risk according to Miettinen’s classification. Preoperative evaluation of risk assessment is required for optimal and personalized treatment planning [8]. 

According to our results, a combined model, based on morphologic and texture features, performed better than models based solely on the two feature classes separately. 

Few recently published manuscripts have already investigated the potential role of radiomics in the risk assessment of GISTs. Chen T. et al. [27] evaluated a radiomic nomogram using morphologic features to predict the malignant risk of GISTs, obtaining an AUC of 0.847 [95% CI: 0.818–0.915], demonstrating that radiomics features combined with clinical data and typical CT characteristics were more effective in evaluating the malignant potential of GISTs compared to clinical data or typical CT characteristics models. Zhang L. et al. [28] also demonstrated favorable performance of a 5-CT-feature-based radiomic model in discriminating risk stratification according to Miettinen’s classification, with an AUC of 0.809 (95% CI: 0.777–0.841). Wang et al. [9] also developed four different radiomic models based on morphological features extracted from arterial and venous enhanced CT scans to predict the malignancy risk of GISTs, which resulted in higher diagnostic performance compared to clinical data and/or typical CT characteristics.

The performance of the combined model obtained in the present study is in line with the performance of previous studies. The main difference between our study and the previous ones is represented by the methodological approach. In particular, inhouse-developed software and complex analyses were used in the previous studies. Such approaches may result in higher performance, thus adversely affecting reproducibility. The latter is a well-known concern for radiomics and AI studies as confirmed by recent initiatives focused on the assessment of quality and reproducibility in this field, such as the Radiomics Quality Scores [29]. In this context, one of the major strengths of our approach is represented by the utilization of a commercially available software. Both TexRAD, for texture feature extraction, and WEKA, for machine learning algorithm development, have been widely utilized and validated, especially for oncologic imaging [14,30,31].

Unlike the previous studies, a reduced number of radiomic features, namely first-level texture features, were included for model building process. Although this might be considered a limitation, it should be noted that in most of the radiomic studies the feature selection process is mandatory in order to avoid overfitting issues.

Another major difference is the use of a machine learning suite (WEKA) with a simple user interface. This software is optimized for supervised machine learning analysis with a specific tool (Auto-WEKA) for feature and classifier optimization. As described in our work, different classifiers and different hyperparameters were used for the three different models. MLP emerged as the best classifier for the morphologic and combined models, even if with different hyperparameters, whereas LWL was the most accurate regarding the texture model. The model training was performed using a 10-fold cross validation method, which allowed us to partially overcome the major limit of the present study, represented by the small sample size. 

Another limitation of the study is the retrospective nature of the patients’ enrollment. However, in line with most of the previous similar studies, this design could be considered appropriate for preliminary results. Finally, one limitation that may be considered is the single slide (2D) tumor manual contouring. In this setting, the use of automatic tumor segmentation software or 3D feature extraction are known to be the best methods. However, the most advanced segmentation tools require specific software, therefore increasing the complexity of the analyses and hampering its implementation in clinical workflow. On the other hand, the 2D approach adopted in our study demonstrated a high reproducibility, which may be well suited for routine usage.

## 5. Conclusions

Noninvasive risk stratification of GISTs may be performed by means of a combined model, based on morphologic and texture features obtained from CT images. The proposed approach, based on commercially available software, might be considered relatively easy to perform and suitable for clinical practice if results can be confirmed in a larger population.

## Figures and Tables

**Figure 1 jpm-13-00717-f001:**
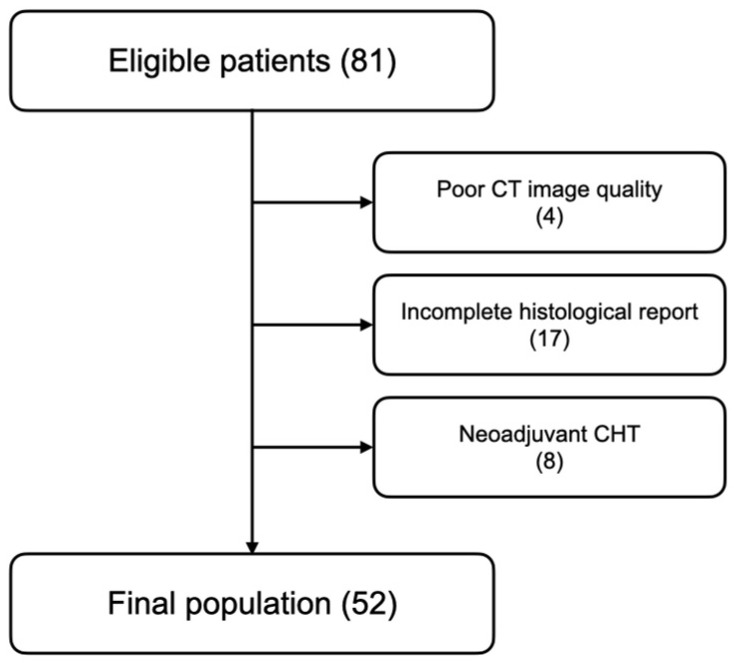
Flow chart detailing the patient selection process.

**Table 1 jpm-13-00717-t001:** Table shows the Miettinen’s risk classification according to GIST location, size and mitotic rate. * HPFs (high power fields).

Size (cm)	Mitotic Rate (HPFs) *	Stomach	Small Bowel	Duodenum	Rectum
≤2	≤5/50	None	None	None	None
>2 ≤ 5	≤5/50	Very low	Low	Low	Low
>5 ≤ 10	≤5/50	Low	Moderate	High	High
>10	≤5/50	Moderate	High		
≤2	>5/50	Insufficient data	Insufficient data	Insufficient data	High
>2 ≤ 5	>5/50	Moderate	High	High	High
>5 ≤ 10	>5/50	High	High	High	High
>10	>5/50	High	High		
≤2	≤5/50	None	None		

**Table 2 jpm-13-00717-t002:** Table shows the subjects characteristics and GIST’s histologic features for the entire population, stratified according to risk and for both training and validation populations.

	All Subjects	Higher Risk	Lower Risk	*p*-Value	Training	Validation	*p*-Value
Subjects	52	22 (42.3%)	30 (57.7%)		35 (67.3%)	17 (32.7%)	
Age	61.64 (±15.09)	64.43 (±14.94)	59.62 (±15.12)	0.2704	62.45 (±15.3)	60.06 (±14.99)	0.3163
Gender (M/F)	31/21	11/11	20/10	0.2262	19/12	11/10	0.5234
Tumor location							
Esophagus	1 (1.9%)	0 (0%)	1 (3.3%)	0.0993	1 (2.9%)	0 (0%)	0.7779
Stomach	37 (71.2%)	17 (77.3%)	20 (66.8%)		23 (65.7%)	14 (80.9%)	
Duodenum	5 (9.6%)	1 (4.5%)	4 (13.3%)		4 (11.4%)	1 (4.8%)	
Jejunum	4 (7.7%)	0 (0%)	4 (13.3%)		3 (8.6%)	1 (4.8%)	
Ileum	5 (9.6%)	4 (18.2%)	1 (3.3%)		4 (11.4%)	1 (9.5%)	
Colon	0 (0%)	0 (0%)	0 (0%)		0 (0%)	0 (0%)	
Tumor size							
≤2	3 (5.8%)	0 (0%)	3 (10%)	<0.0001	2 (5.7%)	1 (5.9%)	0.5037
>2 ≤5	26 (50%)	5 (18.2%)	21 (70%)		20 (57.2%)	6 (35.3%)	
>5 ≤10	11 (21.1%)	5 (27.3%)	6 (20%)		6 (17.1%)	5 (29.4%)	
>10	12 (23,1%)	12 (54.5%)	0 (0%)		7 (20%)	5 (29.4%)	
Mitotic rate							
≤5/50	39 (75%)	9 (40.9%)	30 (100%)	<0.0001	25 (71.4%)	14 (82.3%)	0.3934
>5/50	13 (25%)	13 (59.1%)	0 (0%)		10 (28.6%)	3 (17.7%)	
Risk score							
None	3 (5.8%)	0 (0%)	3 (10%)	<0.0001	2 (5.7%)	1 (5.9%)	0.6809
Very low	14 (27%)	0 (0%)	14 (46.7%)		10 (28.6%)	4 (23.5%)	
Low	13 (25%)	0 (0%)	13 (43.3%)		7 (20%)	6 (35.4%)	
Moderate	11 (21.1%)	11 (50%)	0 (0%)		9 (25.7%)	2 (11.7%)	
High	11 (21.1%)	11 (50%)	0 (0%)		7 (20%)	4 (23.5%)	
Risk Class							
Lower Risk	30 (57.7%)	0 (0%)	30 (100%)	<0.0001	19 (54.3%)	11 (64.7%)	0.4756
Higher Risk	22 (42.3%)	22 (100%)	0 (0%)		16 (45.7%)	6 (35.3%)	

**Table 3 jpm-13-00717-t003:** Table shows the GIST’s morphologic features for the entire population, stratified according to risk and for both training and validation populations.

	All Subjects	Higher Risk	Lower Risk	*p*-Value	Training	Validation	*p*-Value
Subjects	52	22	30		35	17	
Margins							
Regular	41 (78.8%)	12 (54.5%)	29 (96.7%)	0.0002	30 (85.7%)	11 (64.7%)	0.0818
Irregular	11 (21.2%)	10 (45.5%)	1 (3.3%)		5 (14.3%)	6 (35.3%)	
Angiogenesis							
Present	16 (30.8%)	12 (54.5%)	4 (13.3%)	0.0015	10 (28.6%)	6 (35.3%)	0.6222
Absent	36 (69.2%)	10 (45.5%)	26 (86.7%)		25 (71.4%)	11 (64.7%)	
Necrosis							
Present	27 (%)	17 (77.3%)	10 (33.3%)	0.0017	18 (51.4%)	9 (52.9%)	0.9184
Absent	25 (%)	5 (22.7%)	20 (66.7%)		17 (48.6%)	8 (47.1%)	
Peritoneal effusion							
Present	7 (%)	6 (27.3%)	1 (3.3%)	0.0125	4 (11.4%)	3 (17.6%)	0.5377
Absent	45 (%)	16 (72.7%)	29 (96.7%)		31 (88.6%)	14 (82.4%)	
Delta density							
≥55.33	27 (%)	13 (59.1%)	14 (46.7%)	0.3757	19 (54.3%)	6 (35.3%)	0.1985
<55.33	25 (%)	9 (40.9%)	16 (53.3%)		16 (45.7%)	11 (64.7%)	
Peritoneal seeding							
Present	3 (%)	3 (13.6%)	0 (0%)	0.0372	1 (2.8%)	2 (11.8%)	0.1963
Absent	49 (%)	19 (86.4%)	30 (100%)		34 (97.2%)	15 (88.2%)	
Organ invasion							
Present	4 (%)	4 (18.2%)	0 (0%)	0.0151	3 (8.6%)	1 (5.9%)	0.7328
Absent	48 (%)	18 (81.8%)	30 (100%)		32 (91.4%)	16 (94.1%)	
Enhancement pattern							
Homogenous	20 (%)	3 (13.6%)	17 (56.7%)	0.0016	15 (42.8%)	5 (29.4%)	0.3499
Heterogeneous	32 (%)	19 (86.4%)	13 (43.3%)		20 (57.2%)	12 (70.6%)	

**Table 4 jpm-13-00717-t004:** Table shows the GIST’s texture features for the entire population, stratified according to risk, and for both training and validation populations.

	Higher Risk		Lower Risk			Training		Validation		
	Mean	SD	Mean	SD	*p* Value	Mean	SD	Mean	SD	*p* Value
SF0										
Mean	62.36	±26.36	76.13	±28.56	0.0411	68.79	±28.91	73.94	±27.39	0.5853
SD	29.89	±7.27	36.13	±20.12	0.3543	32.48	±13.01	35.57	±21.61	0.9727
Entropy	4.66	±0.21	4.57	±0.35	0.9889	4.59	±0.27	4.64	±0.35	0.4190
MPP	64.90	±24.84	80.05	±27.26	0.0247	71.87	±28	77.29	±25.53	0.5041
Skewness	−0.58	±1.22	−1.63	±2.65	0.8752	−1.09	±2.39	−1.36	±1.81	0.3067
Kurtosis	3.49	±10.51	16.17	±44.37	0.0599	12.14	±41.06	8.06	±15.69	0.5393
SF1										
Mean	4.18	±7.38	12.57	±18.67	0.2135	10.32	±17.6	6.34	±9.45	0.3328
SD	65.50	±19.40	73.19	±33.76	0.2540	68.5	±26.36	72.89	±33.42	0.7317
Entropy	5.45	±0.26	5.31	±0.27	0.9836	5.34	±0.24	5.41	±0.32	0.4178
MPP	53.45	±16.83	58.85	±22.59	0.4228	56.85	±20.46	55.97	±20.71	0.9846
Skewness	0.05	±0.31	0.49	±1.35	0.9480	0.51	±1.11	−0.11	±0.80	0.0533
Kurtosis	0.62	±0.96	5.84	±12.24	0.4386	3.68	±10.35	3.53	±8.17	0.9269
SF1.5										
Mean	8.33	±14.10	19.42	±21.42	0.0997	15.86	±20.24	12.4	±17.64	0.3232
SD	69.11	±19.44	94.64	±51.92	0.0002	84.01	±40.04	83.5	±49.92	0.4506
Entropy	5.48	±0.20	5.43	±0.26	0.9944	5.45	±0.19	5.45	±0.30	0.9989
MPP	57.25	±19.68	73.09	±34.76	0.0188	68.08	±30.42	62.91	±30.2	0.3837
Skewness	0.07	±0.74	0.75	±1.85	0.9196	0.78	±1.53	−0.22	±1.24	0.0312
Kurtosis	2.84	±7.30	7.40	±15.87	0.4984	5.28	±13.94	5.88	±11.39	0.7980
SF1.8										
Mean	11.06	±18.28	25.96	±27.60	0.0271	21.26	±26.26	16.36	±22.64	0.2974
SD	69.32	±20.35	102.90	±61.37	<0.0001	89.43	±47.89	87.19	±58.2	0.2691
Entropy	5.41	±0.28	5.45	±0.28	0.9952	5.43	±0.26	5.44	±0.31	0.8130
MPP	57.78	±22.18	80.07	±41.85	0.0010	73.07	±36.79	65.64	±36.04	0.2526
Skewness	0.14	±1.01	0.82	±1.89	0.9197	0.91	±1.56	−0.25	±1.44	0.0097
Kurtosis	4.47	±10.99	7.16	±14.83	0.6901	5.64	±13.36	6.81	±13.52	0.6533
SF2										
Mean	12.91	±20.94	30.32	±31.85	0.0098	24.84	±30.36	19.07	±25.93	0.2884
SD	64.75	±28.23	107.80	±66.86	<0.0001	89.67	±56.01	89.32	±62.84	0.3449
Entropy	5.45	±0.20	5.46	±0.30	0.9981	5.46	±0.22	5.44	±0.32	0.8290
MPP	58.42	±24.31	84.53	±46.22	0.0001	76.38	±41	67.53	±39.52	0.2710
Skewness	0.19	±1.17	0.84	±1.86	0.9237	0.96	±1.53	−0.25	±1.53	0.0059
Kurtosis	5.45	±13.13	6.78	±13.63	0.8440	5.73	±12.77	7.22	±14.7	0.7025

**Table 5 jpm-13-00717-t005:** Table shows the performances of the three models (Morphologic, Texture and Combined) for the training and validation populations.

	Morphologic	Texture	Combined
**Training**			
Sensitivity	92.31 (63.97–99.81)	75 (50.90–91.34)	100 (78.20–100)
Specificity	81.82 (59.72–94.81)	93.33 (68.05–99.83)	95 (75.13–99.87)
Accuracy	85.71 (69.74–95.19)	82.86 (66.35–93.44)	97.14 (85.08–99.93)
PPV	75 (54.94–88.07)	93.75 (68.95–99.02)	93.75 (68.95–99.02)
NPV	94.74 (73.05–99.17)	73.68 (56.43–85.82)	100 (83.18–100)
AUC	0.848 (0.687–0.946)	0.837 (0.673–0.939)	0.968 (0.846–0.998)
**Validation**			
Sensitivity	66.67 (22.28–95.67)	50 (11.81–88.19)	85.71 (48.69–99.27)
Specificity	81.82 (48.22–97.72)	72.73 (39.03–93.98)	90.91 (62.26–99.53)
Accuracy	76.47 (50.10–93.19)	64.71 (38.33–85.79)	88.89 (65.29–98.62)
PPV	66.67 (33.58–88.78)	50 (22.21–77.79)	85.71 (48.69–99.27)
NPV	81.82 (58.39–93.52)	72.73 (52.56–86.52)	90.91 (62.26–99.53)
AUC	0.742 (0.477–0.919)	0.613 (0.352–0.834)	0.954 (0.752–0.999)

## Data Availability

Data can be made publicly available upon reasonable request.

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
