# Peer review of "Development and Validation of Artificial-Intelligence-Based Radiomics Model Using Computed Tomography Features for Preoperative Risk Stratification of Gastrointestinal Stromal Tumors"

_jpm, 2023, doi:10.3390/jpm13050717_

Round 1

Reviewer 1 Report

Dear Editor,  

Thank you for the opportunity to review this manuscript for the Journal of Personalized Medicine.  

This manuscript reports the results of a study aiming to develop and to validate an AI classification algorithm, based on CT features, to determine high vs low risk patients with GIST tumors. I find it to be a well performed study and a well written manuscript, but I still have a few questions and comments.  

It is stated in the introduction that “…an optimal preoperative evaluation of the tumor is mandatory for an adequate treatment plan” referring the study by Yang et al. However, this study is on how MR can be used to predict mitotic index. It is not about how the preoperative evaluation affects the treatment plan. A better reference could be found to support the claim that an optimal preoperative evaluation is needed for an adequate treatment plan. Furthermore, it would be interesting to know the authors opinion on how the present study could impact the treatment plan. If the results from this study are confirmed in a larger population, could they be directly used in clinical practice in a concrete way or are further research needed?  

The Miettinen’s risk classes where divided into two groups. Why were the cutoff chosen between low risk and moderate risk? What are the clinical implications of this division and how does it affect the use when determining a treatment plan?  

How was the sample size decided?

Author Response

Q1:  "an optimal preoperative evaluation of the tumor is mandatory for an adequate treatment plan” referring the study by Yang et al. However, this study is on how MR can be used to predict mitotic index. It is not about how the preoperative evaluation affects the treatment plan. A better reference could be found to support the claim that an optimal preoperative evaluation is needed for an adequate treatment plan.

R1: Thanks for the suggestion. We updated the reference as requested (Mantese, G. Gastrointestinal Stromal Tumor: Epidemiology, Diagnosis, and Treatment. Curr Opin Gastroenterol 2019, 35, 555–559, doi:10.1097/MOG.0000000000000584)

Q2: Furthermore, it would be interesting to know the authors opinion on how the present study could impact the treatment plan. If the results from this study are confirmed in a larger population, could they be directly used in clinical practice in a concrete way or are further research needed?

R2: Thanks for the interesting question. As for most cancers the preoperative or pretreatment prognosis assessment is extremely important for both therapy personalization and cost-efficiency. The proposed algorithm, if confirmed in larger populations, may be considered as one of the method to establish GIST's prognosis. Moreover, the proposed method can be performed using commercially available free software, thus, ready for clinical practice.

Q3: The Miettinen’s risk classes where divided into two groups. Why were the cutoff chosen between low risk and moderate risk? What are the clinical implications of this division and how does it affect the use when determining a treatment plan?

R3: Thanks for the question. As for the Miettinen's classification and for the NIH consensus classification, the outcome of high and moderate risk is substantially poorer than the other classes. Moreover, the prognosis of the lower risk classes is not markedly inferior of that of the general population (Hsu KH, Yang TM, Shan YS, et al. Tumor size is a major determinant of recurrence in patients with resectable gastrointestinal stromal tumor. Am J Surg 2007;194:148-52.; DeMatteo RP, Gold JS, Saran L, et al. Tumor mitotic rate, size, and location independently predict recurrence after resection of primary gastrointestinal stromal tumor (GIST). Cancer 2008;112:608-15; Steigen SE, Eide RJ. Trends in incidence and survival of mesenchymal neoplasm of the digestive tract within a defined population of northern Norway. APMIS 2006;114:192-200)

Q4: How was the sample size decided?

R4: thanks for the question. For ML analysis the sample size estimation is based on the ratio between the number of features and the input data (patients). The general rule is to use a ratio of 10 (for 10 features you need 100 patients). In this study the higher amount of features analyzed was 38 for the combined model thus for the algorithm development we need 380 input data. However, the AUTO-Weka method, for algorithm training, uses a multiple folds approach that increases proportionally the number of input data and for the combined model we appleid a 12 fold approach, for a total of 420 input data (35 patients of the training population multiplied by 12).

Reviewer 2 Report

 Dear authors, your scientific paper is considered quite interesting, however, this review needs extensive editing. Firstly, I hope that the constructive and thorough examination of your manuscript will help you improve it.  Below you will find my comments/ suggestions:

-        I suggest a slight modification of a title such as:

Development and validation of Artificial- intelligence based radiomics model using computed tomography features for preoperative risk stratification of Gastrointestinal Stromal Tumors

-        The manuscript should be revised at English level by a native speaker. There are several major grammar and vocabulary errors.

-        The abstract is generally comprehensive, slight modifications are recommended such as the extent.

-         

-        Accounting for (replace of with for line 61)

-        Differently from other tumors, GISTs are not staged (incomprehensive, please modify this sentence line 63-64)

-        The introduction needs to be reconstructed: More detailed

You should start with the term, epidemiology, incidence, risk factors, methods for risk assessment. Then, risk stratification and classification methods, prognosis and preoperative pathological examination.

How do you assess the Preoperative pathological examination- risk stratification / significance for GIST patients?

Information about the term Artificial intelligence, future diagnostic-therapeutic perspectives, prognosis.

Term of imaging biomarkers – how you select and assess an imaging biomarker/ pros/cons

Methods/materials

-        More detailed reference about the population in the first paragraph

sex distribution, age, tumor location, tumor size etc. / more detailed exclusion-inclusion criteria

-        What about the expertise of pathologists?

-        Many old references, you should replace them with newer scientific papers.

Author Response

Q1: I suggest a slight modification of a title.

R1: thanks for the suggestion. We modified the title as suggested.

Q2: The manuscript should be revised at English level by a native speaker. There are several major grammar and vocabulary errors.

R2: thanks for the observation. The manuscript as been extensively revised as requested.

Q3: The abstract is generally comprehensive, slight modifications are recommended such as the extent.

R3: thanks for the suggestion. The abstract has been modified as requested.

Q4: Accounting for (replace of with for line 61)

R4: thanks for the observation. The line was modified as requested

Q5: Differently from other tumors, GISTs are not staged (incomprehensive, please modify this sentence line 63-64)

R5: thanks for the suggestion. The manuscript has been modified as requested. "Unlike other tumors, for which the TNM system represent the most commonly adopted staging tool, the risk stratification of GISTs is based on the Miettinen’s classification, which has been recently reviewed..."

Q6: You should start with the term, epidemiology, incidence, risk factors, methods for risk assessment. Then, risk stratification and classification methods, prognosis and preoperative pathological examination.

R6: Thanks for the suggestion. the manuscript has been modified as requested. "These tumors, derive from precursors of interstitial Cajal cells, pacemaker cells responsible for (GI) peristalsis activity. Currently, no environmental risk factor for GIST is known, but there is evidence of familial predisposition to germline oncogene mutations: KIT or PDFRA oncogene mutations are the most frequent [3]. Unlike other tumors, for which the TNM system represent the most commonly adopted staging tool, the risk stratification of GISTs is based on the Miettinen’s classification, which has been recently reviewed [4]. By integrating tumor size (2 cm; >2-5 cm; >5-10 cm, >10 cm), mitotic index (5/50 HPFs or >5/50 HPFs), and tumor site (stomach; duodenum; small bowel; rectum), this classification identifies five risk grades: none, very low, low, moderate, and high. The prognosis of GISTs is closely related to their risk grade. Different risk grades lead to different therapeutical options"

Q7: How do you assess the Preoperative pathological examination- risk stratification/significance for GIST patients?

R7: Thanks for the suggestion. the manuscript has been modified as requested. "Therefore, an adequate preoperative tumor assessment, including specimen collection and pathological examination based on microscopic morphology and immune-phenotype, is mandatory to select an optimal therapeutic strategy for each patient. [5]."

Q8: Information about the term Artificial intelligence, future diagnostic-therapeutic perspectives, prognosis. Term of imaging biomarkers – how you select and assess an imaging biomarker/ pros/cons.

R8: Thanks for the suggestion. the manuscript has been modified as requested.

Q9: More detailed reference about the population in the first paragraph sex distribution, age, tumor location, tumor size etc. / more detailed exclusion-inclusion criteria

R9: thanks for the suggestion. Demographic data are summarized in table 2 including sex distribution, age, tumor location, tumor size for the entire population as well as for the subgroups analyzed (higher and lower risk, training and validation groups).Since this is a retrospective study the inclusion/exclusion criteria are focused on the available data.

Q10: What about the expertise of pathologists?

R10: Thanks for the suggestion. The pathologist was the local expert for GISTs with more than 20 years of practice in the field. The manuscript was updated to include this information.

Q11: Many old references, you should replace them with newer scientific papers.

Q11: thanks for the suggestion. Where possible we updated the references.

Reviewer 3 Report

The article is about “Development and validation of a classification algorithm for  gastrointestinal stromal tumors risk stratification based on  computed tomography features”

Abstract   : well summarized and with basic ideas explained

Intoduction:  Excellent summary of the background and with the reason for carrying out the study clearly explained. Relatively recent bibliography

Material and metohds:  Extensive but adequate explanation due to the complexity of the algorithm to be developed.

Results:  good summary  The tables make it much easier to understand the text.

Discussion: carried out in an orderly manner and with comments on the related articles

The article is about “Development and validation of a classification algorithm for  gastrointestinal stromal tumors risk stratification based on  computed tomography features”

Abstract   : well summarized and with basic ideas explained

Intoduction:  Excellent summary of the background and with the reason for carrying out the study clearly explained. Relatively recent bibliography

Material and metohds:  Extensive but adequate explanation due to the complexity of the algorithm to be developed.

Results:  good summary  The tables make it much easier to understand the text.

Discussion: carried out in an orderly manner and with comments on the related articles

Author Response

Thanks for the comments. We are glad that you appreciate the manuscript